# Photocatalytic Degradation of 1,4-Dioxane by Heterostructured Bi₂O₃/Cu-MOF Composites

**Wen-Min Wang [1], Lu Zhang [2], Wen-Long Wang [1], Jin-Yi Huang [3], Qian-Yuan Wu [1,*] and Jerry J. Wu [3,*]**

[1] State Environmental Protection Key Laboratory of Microorganism Application and Risk Control (SMARC), Guangdong Provincial Engineering Research Center for Urban Water Recycling and Environmental Safety, Institute of Environment and Ecology, Tsinghua Shenzhen International Graduate School, Tsinghua University, Shenzhen 518055, China; wangwm21@mails.tsinghua.edu.cn (W.-M.W.); wwl20@sz.tsinghua.edu.cn (W.-L.W.)

[2] Key Laboratory of Groundwater Resources and Environment, Ministry of Education, Jilin University, Changchun 130026, China; lzhang22@mails.jlu.edu.cn

[3] Department of Environmental Engineering and Science, Feng Chia University, Taichung 407, Taiwan; aa7483569@gmail.com

* Correspondence: wu.qianyuan@sz.tsinghua.edu.cn (Q.-Y.W.); jjwu@mail.fcu.edu.tw (J.J.W.); Tel.: +86-0755-2603-6701 (Q.-Y.W.); +886-4-24517250 (ext. 5206) (J.J.W.)

**Abstract:** Photocatalysts exhibiting high activity for the degradation of 1,4-dioxane (1,4-D) have been a subject of intense focus due to their high toxicity and challenging degradability. Bismuth oxide ($Bi_2O_3$) is recognized as an ideal photocatalyst; however, there have been limited studies on its effectiveness in 1,4-D degradation. It is crucial to address the issue of low photocatalytic efficiency attributed to the instability and easy recombination of photogenerated electrons and holes in $Bi_2O_3$ upon photoexcitation. In this study, Cu-MOF and oxygen vacancy were utilized to improve the 1,4-D photocatalytic degradation efficiency of $Bi_2O_3$ by preparing $Bi_2O_3$, $Bi_2O_3$/Cu-MOF, $Bi_2O_{3-x}$, and $Bi_2O_{3-x}$/Cu-MOF. The results revealed that the incorporation of Cu-MOF induced a larger specific surface area, a well-developed pore structure, and a smaller particle size in $Bi_2O_3$, facilitating enhanced visible light utilization and an improved photoelectron transfer rate, leading to the highest photocatalytic activity observed in $Bi_2O_3$/Cu-MOF. In addition, oxygen vacancies were found to negatively affect the photocatalytic activity of $Bi_2O_3$, mainly due to the transformation of the $\beta$-$Bi_2O_3$ crystalline phase into $\alpha$-$Bi_2O_3$ caused by oxygen vacancies. Further, the synergistic effect of MOF and oxygen vacancies did not positively affect the photocatalytic activity of $Bi_2O_3$. Therefore, the construction of heterojunctions using Cu-MOF can significantly enhance the efficiency of degradation of 1,4-D, and $Bi_2O_3$/Cu-MOF appears to be a promising photocatalyst for 1,4-D degradation. This study opens new avenues for the design and optimization of advanced photocatalytic materials with improved efficiency for the treatment of recalcitrant organic pollutants.

**Keywords:** bismuth oxide; Cu-MOF; oxygen vacancy; photocatalysis

## 1. Introduction

1,4-dioxane (1,4-D) has found widespread use in various industrial applications, such as textiles, dyes, pharmaceuticals, pesticides, varnishes, degreasers, and cosmetics, as an industrial solvent stabilizer. Due to its high solubility and low volatility, 1,4-D exhibits remarkable stability, making it a significant threat once released into water bodies and soil through industrial wastewater discharge. More importantly, it poses a considerable risk to human health, as research has indicated a link between 1,4-D exposure and an increased likelihood of cancer, including breast cancer and endometrial cancer [1]. In addition, it negatively impacts certain human sex hormones. 1,4-D has been recognized as a Class 2B carcinogen by the US Environmental Protection Agency [2]. Efforts to mitigate its presence have encountered challenges, with current studies highlighting the inefficiency

of biological treatment due to 1,4-D's resistance to biodegradation. Traditional physical methods (e.g., air stripping, carbon adsorption) may also prove inadequate in removing 1,4-D. Although the most effective distillation method exists, it is not widely adopted because of its high cost [3]. Advanced oxidation processes (AOPs) have been considered a promising alternative for 1,4-D degradation, exhibiting significant efficiency [4]. Among these, solar-driven photocatalytic techniques have garnered increasing attention as they offer lower process costs and avoid the utilization of oxidants [5]. However, for the successful degradation of 1,4-D, the development of photocatalysts with high catalytic activity remains essential.

Conventional photocatalysts, such as $TiO_2$, suffer from poor solar utilization efficiency due to their broad band gap [6,7]. In contrast, $Bi_2O_3$ offers several advantages, including controllable morphologies ($\alpha$-$Bi_2O_3$, $\beta$-$Bi_2O_3$, and $\gamma$-$Bi_2O_3$), a narrow band gap ($\sim$2.80 eV), and good electrical and thermal conductivity, making it an ideal photocatalytic material [8,9]. Nevertheless, the degradation efficiency of 1,4-D by $Bi_2O_3$ under solar light remains unclear. Additionally, the unstable structure of $Bi_2O_3$ compounds and the ease of recombination of photogenerated electrons and holes contribute to its low photocatalytic efficiency. Various modification methods have been attempted to improve the photocatalytic activity of $Bi_2O_3$. One effective approach for $Bi_2O_3$ modification involves constructing heterostructures, leading to the development of novel photocatalytic materials, such as $Bi_2O_3$/$TiO_2$ [10], $Bi_2O_3$/g-$C_3N_4$ [11], $Bi_2O_3$/$Bi_2O_2CO_3$ [12], and $Bi_2O_3$/$Bi_2S_3$/$MoS_2$ [13]. These heterostructures effectively modulate the energy band structure and facilitate electron transfer at the interface. At the same time, they increase the specific surface area of the material and expose more active sites, thereby improving the overall photocatalytic activity [14].

Metal organic frameworks (MOFs) are polycrystalline coordination polymers with a multidimensional stereospecific structure, formed by combining metal ions (or metal oxide clusters) with organic ligand molecules through coordination bonds. They have the advantages of high specific surface area, structural stability, and adjustable pore size, making them highly attractive for applications in photocatalysis, gas separation, and storage [15]. The use of MOF materials in constructing heterogeneous structures can significantly enhance the photocatalytic activity of catalysts. For example, Akbarzadeh et al. [16] developed the Cu-MOF/rGO/$Ag_3VO_4$ for photocatalytic degradation of AB92 dye with a 3-h degradation efficiency of 93.75%. Moreover, vacancy engineering, especially focusing on oxygen vacancies, has made great progress in the field of photocatalysis. In general, oxygen vacancies in photocatalysts can effectively improve light absorption and promote charge separation efficiency by creating unsaturated sites on the surface [17]. Ding et al. [18] successfully prepared HS-$CuFe_2O_4$-$\sigma$ with abundant oxygen vacancies, leading to a remarkable 20-fold increase in the rate of ciprofloxacin degradation at neutral pH. Similarly, Zou et al. [19] designed MOF-derived $Bi_2O_3$@C with abundant oxygen vacancies, demonstrating highly efficient photodegradation of tetracycline hydrochloride with an 88% removal rate within just 120 min. These findings suggest that the utilization of MOFs in constructing heterostructures or inducing oxygen vacancies may hold great potential in enhancing the photocatalytic degradation of 1,4-D. However, few studies have been conducted on the individual and synergistic effects of MOF-constructed heterostructures and oxygen vacancies on the 1,4-D degradation by $Bi_2O_3$. Therefore, exploring these aspects could prove to be significant for advancing the field of photocatalysis.

This study aimed to address the research gap concerning the photocatalytic degradation of 1,4-D by $Bi_2O_3$. To achieve this, the individual contributions of MOF-constructed heterojunctions and oxygen vacancy engineering to enhancing the photocatalytic activity of $Bi_2O_3$ in the degradation of 1,4-D were investigated. Furthermore, the study explored the synergistic effects resulting from the combination of MOF-constructed heterojunctions and oxygen vacancies, providing valuable insights into the overall catalytic mechanism. For this purpose, $Bi_2O_3$-related photocatalysts were prepared using a simple microwave method to degrade 1,4-D. Four photocatalysts, namely, $Bi_2O_3$, $Bi_2O_3$/Cu-MOF, $Bi_2O_{3-x}$,

and $Bi_2O_{3-x}$/Cu-MOF, were synthesized to explore the effects of MOF-constructed heterostructures and oxygen vacancies on the photocatalytic activity. The factors and mechanism that enhance the photocatalytic activity of $Bi_2O_3$ were elucidated by analyzing the characterization results and optical properties of these as-prepared photocatalysts. In addition, degradation experiments were conducted to determine the 1,4-D degradation efficiency of the above-mentioned $Bi_2O_3$-based photocatalysts.

## 2. Results and Discussion

### 2.1. Characterizations

#### 2.1.1. Morphology and Structure

The morphology and microstructure of the samples were observed through FE-SEM and TEM analysis. Cu-MOF displayed an octahedral structure, typical of HKUST-1 MOF crystals, confirming the successful synthesis of Cu-MOF, as shown in Figure 1 [20]. $Bi_2O_3$ presented a sphere-like structure with an average diameter of about 1–2 μm (Figure 2a). Further examination at high magnification revealed that the peripheries of microsphere formations were constructed from abundant ultrathin nanosheets, each measuring approximately 10 nm in thickness. $Bi_2O_3$/Cu-MOF microsphere showed a more advanced porous structure compared to $Bi_2O_3$ (Figure 2b). This porous structure was conducive to increasing the active sites and promoting interfacial charge transfer, which was beneficial for the photodegradation [21]. With the introduction of oxygen vacancy, the $Bi_2O_{3-x}$ microsphere agglomerated (Figure 2c). The nanosheets on the surface of $Bi_2O_{3-x}$ microspheres collapsed and sporadically attached irregular particles, potentially obstructing the active sites. $Bi_2O_{3-x}$/Cu-MOF exhibited an irregular shape, and the nanosheets on its surface disappeared (Figure 2d). The low-magnification TEM images corroborated the SEM results. It is noteworthy that the TEM image of $Bi_2O_3$/Cu-MOF also showcased the presence of more advanced pore structures.

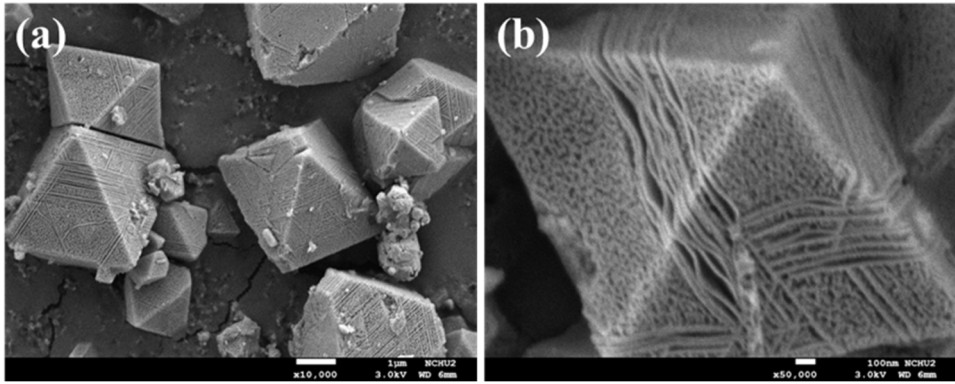

**Figure 1.** FE-SEM images of Cu-MOF with different magnitudes of (**a**) ×10,000 and (**b**) ×50,000.

#### 2.1.2. XRD Analysis

XRD analysis was performed to characterize the crystallinity and crystal phase of Cu-MOF, $Bi_2O_3$, $Bi_2O_3$/Cu-MOF, $Bi_2O_{3-x}$, and $Bi_2O_{3-x}$/Cu-MOF photocatalysts, as shown in Figure 3. The XRD pattern of Cu-MOF matched well with the simulated pattern of $Cu_3(BTC)_2 \cdot 3H_2O$ (also known as HKUST-1) [22]. The main peaks of Cu-MOF were located at 6.7°, 9.46°, 11.62°, 13.42°, 14.62°, 16.42°, 17.44°, 19.02°, 20.1°, 21.24°, 23.38°, 24.06°, 25.94°, 28.66°, and 29.32°, which correspond to the (200), (220), (222), (400), (331), (422), (511), (440), (600), (620), (444), (551), (731), (733), and (751) planes, respectively (JCPDS card No. 00-062-1183) [23,24].

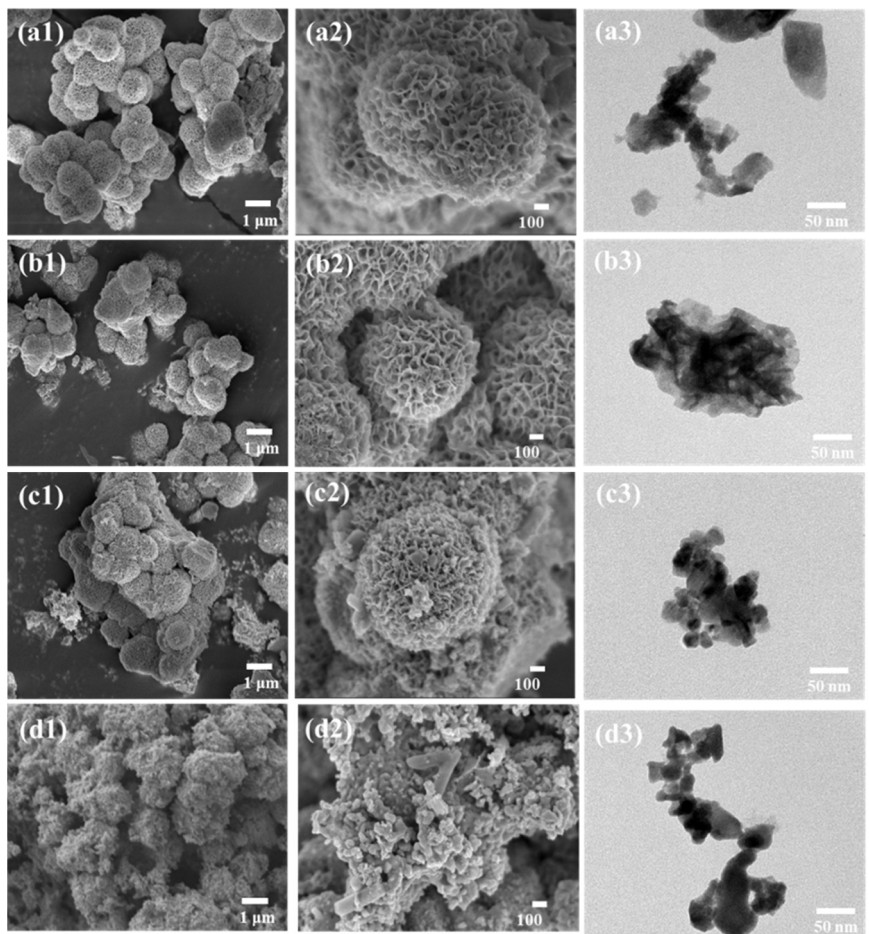

**Figure 2.** FE-SEM ((**a1,a2,b1,b2,c1,c2,d1,d2**) with different magnitudes) and TEM images (**a3,b3,c3,d3**) of (**a**) $Bi_2O_3$, (**b**) $Bi_2O_3/Cu\text{-}MOF$, (**c**) $Bi_2O_{3-x}$, (**d**) $Bi_2O_{3-x}/Cu\text{-}MOF$.

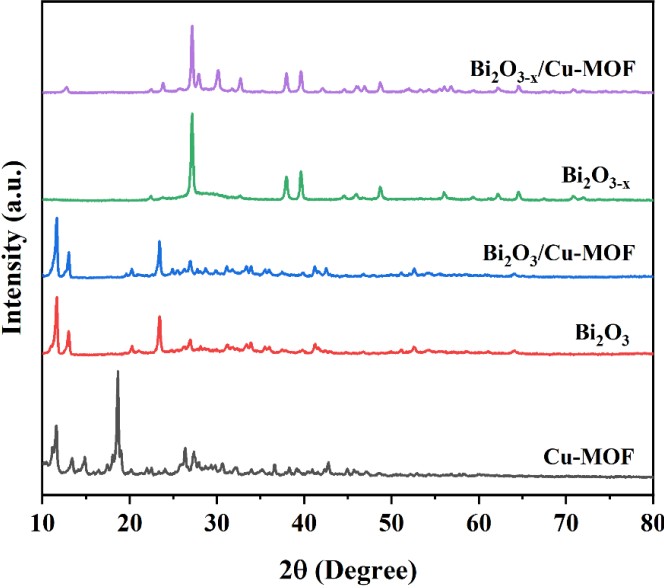

**Figure 3.** XRD patterns of Cu-MOF, $Bi_2O_3$, $Bi_2O_3/Cu\text{-}MOF$, $Bi_2O_{3-x}$, and $Bi_2O_{3-x}/Cu\text{-}MOF$.

There were no significant changes in the characteristic peaks of $Bi_2O_3$ and $Bi_2O_{3-x}$ after Cu-MOF doping, indicating that the addition of Cu-MOF had negligible effects on the crystalline structure of the $Bi_2O_3/Cu\text{-}MOF$ and $Bi_2O_{3-x}/Cu\text{-}MOF$ composite photo-

catalysts [25]. The main diffraction peaks of $Bi_2O_3$ and $Bi_2O_3/Cu$-MOF were located at 2θ values of 10.07°, 23.22°, and 26.38°, which could be well indexed to the (004), (101), and (105) planes, respectively, of β-$Bi_2O_3$ (JCPDS 76-2477) [26]. Both $Bi_2O_3$ and $Bi_2O_3/Cu$-MOF exhibited metastable phase β-$Bi_2O_3$ due to the microwave synthesis method, which promoted the formation of metastable phases by controlling the reaction kinetics [27]. The main diffraction peaks of $Bi_2O_{3-x}$ and $Bi_2O_{3-x}/Cu$-MOF were located at 2θ values of 26.92°, 37.60°, 40.05°, and 48.58°, which could be well indexed to the (111), (112), (−222), and (−104) planes, respectively, of α-$Bi_2O_3$ (PDF NO. 41-1449) [28]. The generation of oxygen vacancies transformed metastable phase β-$Bi_2O_3$ into the stable α-$Bi_2O_3$, indicating that oxygen vacancy significantly influenced the crystal structure of the catalyst.

### 2.1.3. XPS Analysis

X-ray photoelectron spectroscopy (XPS) was employed to determine the chemical compositions and surface chemical states of $Bi_2O_3$, $Bi_2O_3/Cu$-MOF, $Bi_2O_{3-x}$, and $Bi_2O_{3-x}/Cu$-MOF, as shown in Figure 4. The surface composition elements of the four photocatalysts were detected as Bi and O (Figure 4a).

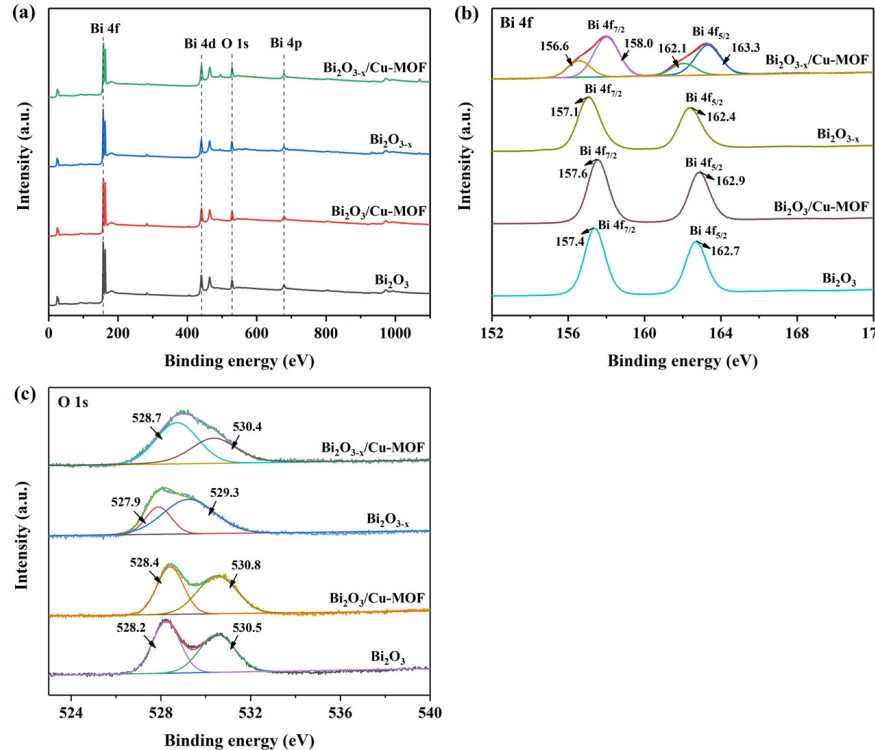

**Figure 4.** XPS spectra of (**a**) a wide survey, (**b**) Bi 4f, and (**c**) O 1s over $Bi_2O_3$, $Bi_2O_3/Cu$-MOF, $Bi_2O_{3-x}$, and $Bi_2O_{3-x}/Cu$-MOF.

Figure 4b shows the high-resolution XPS spectra of Bi element. The $Bi_2O_3$ spectrum exhibited two characteristic peaks located at 157.4 eV and 162.7 eV, which should be assigned to Bi $4f_{5/2}$ and Bi $4f_{7/2}$ regions of $Bi^{3+}$ species, respectively [29]. When Cu-MOF was introduced, the characteristic peaks of Bi shifted to higher binding energies (157.6 eV and 162.9 eV), suggesting that the Cu-MOF introduction induced the electron transfer of Bi to the surrounding adjacent atoms, leading to a more positive valence state than $Bi^{3+}$ species of the Bi element in $Bi_2O_3/Cu$-MOF. In contrast, the characteristic peaks of Bi in $Bi_2O_{3-x}$ shifted to lower binding energies (157.1 eV and 162.4 eV), which should be ascribed to the presence of oxygen vacancies [19]. Furthermore, the peaks of Bi $4f_{5/2}$ and Bi $4f_{7/2}$ of $Bi_2O_{3-x}/Cu$-MOF could be subdivided into two peaks, respectively, probably due to the combined effect of MOF and oxygen vacancies. It can be deduced that most of the Bi element was affected by the Cu-MOF, with peaks shifting towards higher binding energies

(158.0 eV and 163.3 eV), while a small proportion shifted to lower binding energies due to the presence of oxygen vacancies (156.6 eV and 162.1 eV).

Figure 4c shows the high-resolution XPS spectra of the O element. In the spectrum of $Bi_2O_3$, the peaks at 528.2 eV and 530.5 eV should be attributed to the Bi−O bond and adsorbed oxygen, respectively [30]. The corresponding peaks of the O element in $Bi_2O_3$/Cu-MOF shifted to higher binding energies (528.4 eV and 530.8 eV). This shift could be attributed to the rebalancing of inner electrons, proving a strong interfacial interaction between the $Bi_2O_3$ and Cu-MOF heterojunction, which was beneficial for the separation and transfer of the photogenerated charge carrier [31]. $Bi_2O_{3-x}$ appeared as two peaks located at 527.9 eV and 529.3 eV, which should be attributed to the Bi−O bond and oxygen vacancy [32]. These two peaks also shifted towards higher binding energies in $Bi_2O_{3-x}$/Cu-MOF, consistent with the shift of O 1s in $Bi_2O_3$/Cu-MOF. Therefore, the introduction of Cu-MOF usually resulted in a positive shift of the O1s peak. Moreover, it was noteworthy that the peak intensity of the oxygen vacancy in $Bi_2O_{3-x}$/Cu-MOF was much lower than that in $Bi_2O_{3-x}$. This phenomenon indicated that the introduction of Cu-MOF led to the generation of even fewer oxygen vacancies.

### 2.1.4. BET and Particle Size Analysis

The BET-specific surface area ($S_{BET}$) of photocatalysts was determined through the $N_2$ adsorption–desorption experiment, and the results are shown in Table 1. Upon the introduction of Cu-MOF, the specific surface area of $Bi_2O_3$ increased from 5.09 $m^2/g$ ($Bi_2O_3$) to 9.52 $m^2/g$ ($Bi_2O_3$/Cu-MOF), and that of $Bi_2O_{3-x}$ increased from 6.08 $m^2/g$ ($Bi_2O_{3-x}$) to 7.81 $m^2/g$ ($Bi_2O_{3-x}$/Cu-MOF). The favorable increase in the specific surface area might be due to its framework structure, which is consistent with previous findings [19]. At the same time, the presence of oxygen vacancies also contributed to the increase in the specific surface area. The expanded surface area allowed for more active sites, facilitating easy transport of charge carriers, and ultimately enhancing the photocatalytic activity [17,33]. Notably, $Bi_2O_3$/Cu-MOF exhibited the highest $S_{BET}$ value (9.52 $m^2/g$) among the prepared photocatalysts, in agreement with the porous structure observed by SEM.

**Table 1.** $S_{BET}$ of $Bi_2O_3$, $Bi_2O_3$/Cu-MOF, $Bi_2O_{3-x}$, and $Bi_2O_{3-x}$/Cu-MOF.

| Photocatalyst | $Bi_2O_3$ | $Bi_2O_3$/Cu-MOF | $Bi_2O_{3-x}$ | $Bi_2O_{3-x}$/Cu-MOF |
|---|---|---|---|---|
| $S_{BET}$ ($m^2/g$) | 5.09 | 9.52 | 6.08 | 7.81 |

The particle size distribution of the photocatalysts was measured using a laser particle size analyzer, as shown in Figure 5. The median particle diameter ($D_{50}$) of $Bi_2O_3$/Cu-MOF (3.89 μm) was smaller than that of $Bi_2O_3$ (4.39 μm) and the $D_{50}$ of $Bi_2O_{3-x}$/Cu-MOF (2.63 μm) was smaller than that of $Bi_2O_{3-x}$ (4.32 μm). These data indicated that the introduction of Cu-MOF was beneficial in reducing the particle size, which could promote photocatalytic activity [34]. Moreover, the $D_{50}$ of $Bi_2O_{3-x}$ was smaller than that of $Bi_2O_3$ and the $D_{50}$ of $Bi_2O_{3-x}$/Cu-MOF was smaller than that of $Bi_2O_3$/Cu-MOF. This result could be attributed to the oxygen vacancy promoting the conversion of crystal from β-$Bi_2O_3$ to α-$Bi_2O_3$ with a smaller particle size.

### 2.1.5. UV-Vis DRS Analysis

The UV-Vis DRS of photocatalysts is shown in Figure 6a. All photocatalysts exhibited strong absorption in the UV region (220–300 nm). Specifically, $Bi_2O_3$ displayed the highest absorption intensity in the UV region (220–350 nm) but the weakest absorption intensity in the visible region (400–800 nm). Upon the introduction of Cu-MOF, the absorption intensity of $Bi_2O_3$/Cu-MOF and $Bi_2O_{3-x}$/Cu-MOF decreased in the UV region, while it increased in the visible region. This change could be attributed to the unique d–d transition of the Cu ions in Cu-MOF and the strong interaction between $Bi_2O_3$ or $Bi_2O_{3-x}$

and Cu-MOF [35,36]. The enhanced light absorption in the visible region could provide more photogenerated charge carriers, leading to enhanced photocatalytic activity [37]. In addition, it was observed that the absorption intensity of $Bi_2O_{3-x}$ was higher than that of $Bi_2O_3$, and $Bi_2O_{3-x}/Cu$-MOF exhibited higher absorption intensity than $Bi_2O_3/Cu$-MOF in the visible region. This phenomenon elucidated that the presence of oxygen vacancies could enhance the absorption intensity within the visible spectral range. The absorption peaks of Cu-MOF, $Bi_2O_3/Cu$-MOF, and $Bi_2O_{3-x}/Cu$-MOF at approximately 700 nm might be caused by the spin-enabled transition of the material [33]. Therefore, the introduction of Cu-MOF and oxygen vacancies had the potential to enhance the utilization efficiency of visible light.

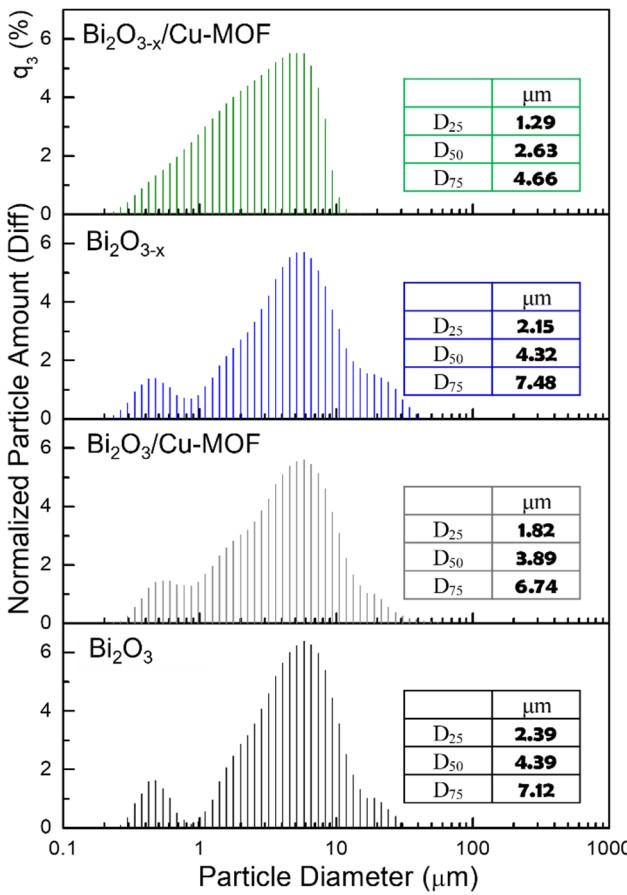

**Figure 5.** Particle size distribution of $Bi_2O_3$, $Bi_2O_3/Cu$-MOF, $Bi_2O_{3-x}$, and $Bi_2O_{3-x}/Cu$-MOF.

Based on UV-Vis DRS spectra and Tauc's approach (Equation (1)) [38], the bandgap ($E_g$) values of photocatalysts were calculated and are presented in Figure 6b and Table 2. Obviously, the band gap of $Bi_2O_3/Cu$-MOF (3.30 eV) was smaller than that of $Bi_2O_3$ (3.54 eV), and that of $Bi_2O_{3-x}/Cu$-MOF (2.19 eV) was smaller than that of $Bi_2O_{3-x}$ (3.30 eV). The reduction in the band gaps of $Bi_2O_3/Cu$-MOF and $Bi_2O_{3-x}/Cu$-MOF was mainly attributed to the heterojunction formed between $Bi_2O_3$ or $Bi_2O_{3-x}$ and Cu-MOF [17]. The narrow band gap allows electrons to be excited from the valence band (VB) to the conduction band (CB) with less energy, which facilitates the separation and transfer of photogenerated electrons and holes and improves photocatalytic activity [34,39]. Moreover, the band gap of $Bi_2O_{3-x}$ (3.30 eV) was smaller than that of $Bi_2O_3$ (3.54 eV), and the band gap of $Bi_2O_{3-x}/Cu$-MOF (2.19 eV) was smaller than that of $Bi_2O_3/Cu$-MOF (3.30 eV). These results demonstrated that oxygen vacancies could effectively reduce the band gap, consistent with the findings of Zhao et al. [17].

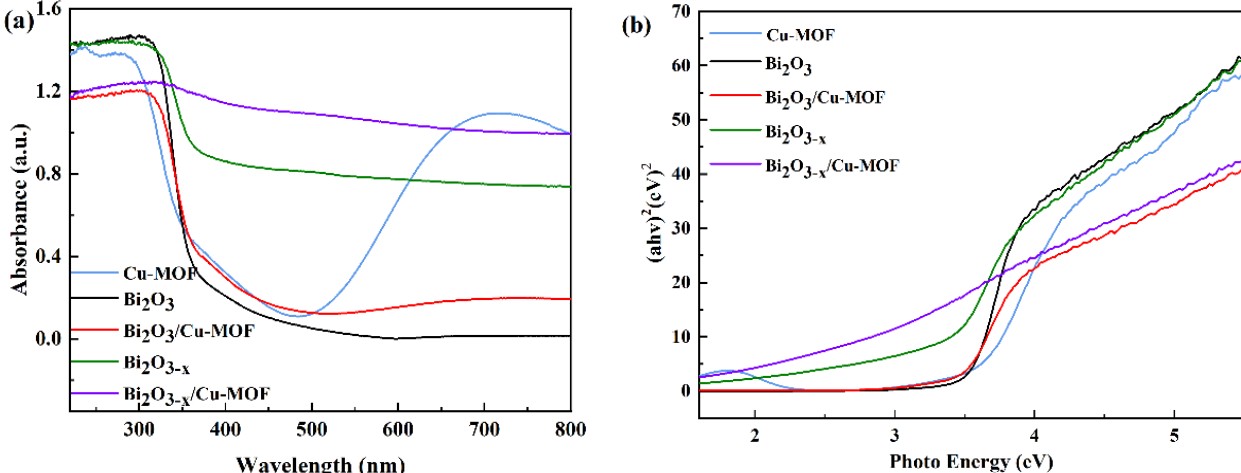

**Figure 6.** (**a**) UV-Vis absorption spectra and (**b**) Tauc's plots of Cu-MOF, $Bi_2O_3$, $Bi_2O_3$/Cu-MOF, $Bi_2O_{3-x}$, and $Bi_2O_{3-x}$/Cu-MOF.

$$[(\alpha h\upsilon)^{1/n} = A(h\upsilon - E_g)] \tag{1}$$

where $\alpha$ is the absorption coefficient, hv is the photo energy, A is a constant, and n is either 1/2 for a direct transition or 2 for an indirect transition.

**Table 2.** Band gap energies of Cu-MOF, $Bi_2O_3$, $Bi_2O_3$/Cu-MOF, $Bi_2O_{3-x}$, and $Bi_2O_{3-x}$/Cu-MOF.

| Photocatalyst | Band Gap (eV) |
|:---:|:---:|
| Cu-MOF | 3.59 |
| $Bi_2O_3$ | 3.54 |
| $Bi_2O_3$/Cu-MOF | 3.46 |
| $Bi_2O_{3-x}$ | 3.30 |
| $Bi_2O_{3-x}$/Cu-MOF | 2.19 |

### 2.1.6. PL Spectra Analysis

Fluorescence emission detection (PL) was employed to determine the charge generation rate and the recombination of photo-generated electrons and holes. A low fluorescence intensity suggested a low recombination rate and a high electron transfer efficiency, leading to an improvement in the photocatalytic activity. The PL spectra of $Bi_2O_3$, $Bi_2O_3$/Cu-MOF, $Bi_2O_{3-x}$, $Bi_2O_{3-x}$/Cu-MOF, and Cu-MOF are shown in Figure 7. Cu-MOF showed a low-level peak intensity within the wavelength range of 400~800 nm, indicating its low recombination rate and higher electron transfer efficiency. Distinctive peaks were observed around 649.8 nm and 818.7 nm in $Bi_2O_3$, $Bi_2O_3$/Cu-MOF, $Bi_2O_{3-x}$, and $Bi_2O_{3-x}$/Cu-MOF, with the peak intensity following the order of $Bi_2O_3$ > $Bi_2O_3$/Cu-MOF > $Bi_2O_{3-x}$ > $Bi_2O_{3-x}$/Cu-MOF. This trend suggested that the introduction of Cu-MOF reduced the recombination of electrons and holes due to the lower peak intensity observed in $Bi_2O_3$/Cu-MOF compared to $Bi_2O_3$ and in $Bi_2O_{3-x}$/Cu-MOF compared to $Bi_2O_{3-x}$. This effect was attributed to the presence of heterojunctions. Furthermore, the peak intensity of $Bi_2O_{3-x}$ was lower than that of $Bi_2O_3$, and the peak of $Bi_2O_{3-x}$/Cu-MOF was lower than that of $Bi_2O_3$/Cu-MOF, suggesting that the presence of oxygen vacancies significantly enhanced the separation efficiency of photoinduced carriers and reduced the recombination of electrons and holes. Therefore, both heterojunctions and oxygen vacancies proved to be effective in reducing the recombination efficiency of electron holes.

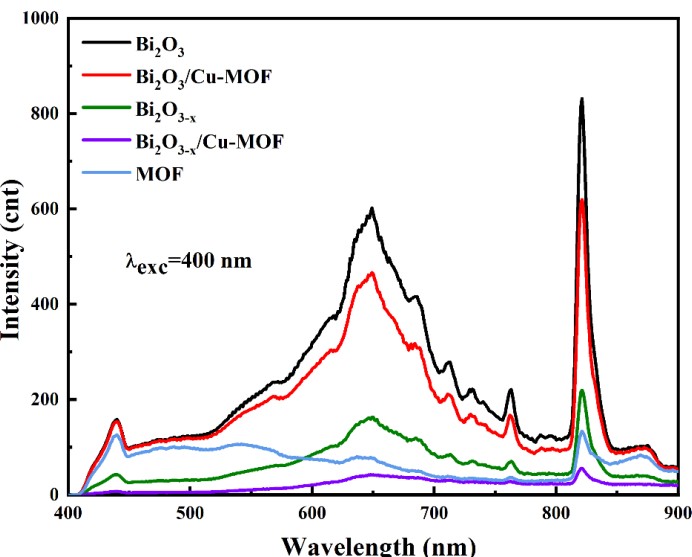

**Figure 7.** PL spectra of as-synthesized Cu-MOF, $Bi_2O_3$, $Bi_2O_3/Cu$-MOF, $Bi_2O_{3-x}$, and $Bi_2O_{3-x}/Cu$-MOF photocatalysts.

## 2.2. Photocatalytic Activity Performance

To evaluate the effect of heterojunction construction and oxygen vacancy on the performance of the photocatalyst, experiments were conducted for photocatalytic degradation of 1,4-D using the prepared catalysts. Figure 8a shows that the removal efficiency of 1,4-D during the dark adsorption–desorption phase was negligible, indicating minimal adsorption of 1,4-D onto the surfaces of the photocatalysts. During the photocatalytic stage, the 1,4-D removal efficiency followed the order of $Bi_2O_3/Cu$-MOF (68.7%) > $Bi_2O_3$ (62.8%) > $Bi_2O_{3-x}/Cu$-MOF (52.2%) > $Bi_2O_{3-x}$ (47.1%) within 180 min, with the kinetic rate constant ($k_{obs}$) values following the same order (Figure 8b). Table 3 shows the 1,4-D removal efficiency of different photocatalysts. It was found that $Bi_2O_3/Cu$-MOF exhibited higher photocatalytic activity for 1,4-D degradation compared to most of those previously reported. The enhanced photocatalytic activity of $Bi_2O_3/Cu$-MOF could be attributed to several factors: (i) the advanced porous structure, large specific surface area, and small particle size providing a large number of active sites; (ii) the heterojunction promoting visible light utilization, electron transfer rate, and efficient separation of photogenerated electron–hole pairs [40]; and (iii) the catalytic activity of the crystal structure of β-$Bi_2O_3$ ($Bi_2O_3$ and $Bi_2O_3/Cu$-MOF) being higher than that of α-$Bi_2O_3$ ($Bi_2O_{3-x}$ and $Bi_2O_{3-x}/Cu$-MOF) [28].

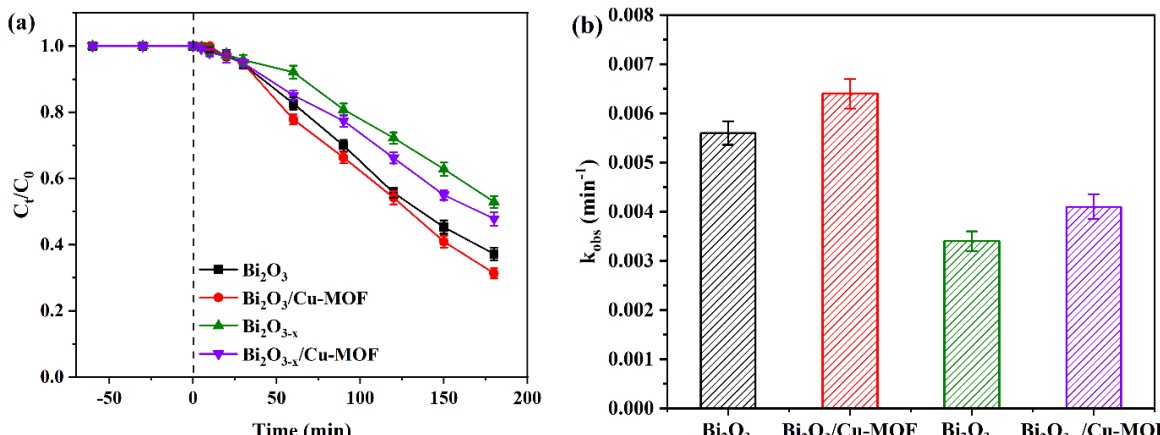

**Figure 8.** Photocatalytic activities of $Bi_2O_3$, $Bi_2O_3/Cu$-MOF, $Bi_2O_{3-x}$, and $Bi_2O_{3-x}/Cu$-MOF: (**a**) removal efficiency of 1,4-D and (**b**) reaction rate constant ($k_{obs}$).

**Table 3.** Comparison of photocatalytic activities of photocatalysts for 1,4-D removal.

| Photocatalyst | Light Source | Catalyst Dosage (g/L) | 1,4-D Concentration (mg/L) | Time (min) | 1,4-D Removal Efficiency (%) | References |
|---|---|---|---|---|---|---|
| Fe/nAl | Solar | 0.3 | 50 | 180 | about 22 | [5] |
| $WO_3/n\gamma$-$Al_2O_3$ | Solar | 0.3 | 50 | 180 | 56.67 | [41] |
| $TiO_2$ | Xenon light (2 kW) | 0.5 | 500 | 180 | about 10 | [42] |
| Au–$TiO_2$ | Xenon light (2 kW) | 0.5 | 500 | 240 | 59 | [43] |
| Cu-ZnO | Solar | 0.3 | 355 | 180 | 43.9 | [44] |
| $Bi_2O_3$/Cu-MOF | Xenon light (350 W) | 0.06 | 50 | 180 | 68.7 | This study |

Despite $Bi_2O_{3-x}$ and $Bi_2O_{3-x}$/Cu-MOF with oxygen vacancies exhibiting excellent optical properties, such as strong adsorption in the visible region, a small band gap, and weak PL peak intensity, their photocatalytic activity was inferior compared to that of $Bi_2O_3$ and $Bi_2O_3$/Cu-MOF. This difference may be attributed to the following reasons: (i) the crystal structure of $\beta$-$Bi_2O_3$ ($Bi_2O_3$ and $Bi_2O_3$/Cu-MOF) having higher catalytic activity than $\alpha$-$Bi_2O_3$ ($Bi_2O_{3-x}$ and $Bi_2O_{3-x}$/Cu-MOF) [28]; (ii) the oxygen vacancy structure leaving two electrons due to the charge compensation effect, occupying the position of photoexcited electrons, which leads to a decrease in the number of oxidative holes ($h^+$) [18]; and (iii) the introduction of oxygen vacancies lowering the zero charge points of $Bi_2O_{3-x}$ and $Bi_2O_{3-x}$/Cu-MOF below the solution pH value, thereby inhibiting the degradation of 1,4-D due to its negatively charged-surface [4].

In summary, heterojunctions can significantly improve the structure and optical properties of the materials, while promoting photocatalytic activity. On the other hand, although oxygen vacancies could improve the optical properties of the materials, they are not conducive to the oxidation reaction in the photocatalytic system.

### 2.3. Photocatalytic Mechanisms

Based on the above discussion, $Bi_2O_3$/MOF was a composite of $\beta$-$Bi_2O_3$ and $Cu_3(BTC)_2 \bullet 3H_2O$ (Cu-MOF). The values of VB and CB of $\beta$-$Bi_2O_3$ and Cu-MOF were calculated using Equations (1) and (2) [45]. The X values of $\beta$-$Bi_2O_3$ and Cu-MOF were 6.12 eV and 5.72 eV [45,46]. Through calculations, it was determined that the $E_{VB}$ and $E_{CB}$ of $\beta$-$Bi_2O_3$ are 3.39 eV and $-0.15$ eV, and the $E_{VB}$ and $E_{CB}$ of Cu-MOF are about 3.02 eV and $-0.57$ eV.

$$E_{VB} = X - E_e + 0.5E_g \tag{2}$$

$$E_{CB} = E_{VB} - E_g \tag{3}$$

where $E_g$ is the band gap of the semiconductor, $E_{VB}$ is the VB edge potential, $E_{CB}$ is the CB edge potential, $E_e$ is the free electron energy (4.5 eV) on the NHE (pH = 7), and X is the electronegativity of the semiconductor.

As reported, $\beta$-$Bi_2O_3$ and Cu-MOF are n-type semiconductors with a Fermi energy level ($E_f$) close to its CB [9,46], forming an n-n heterojunction between $\beta$-$Bi_2O_3$ and Cu-MOF. Since the $E_f$ of Cu-MOF is located at more negative potential than $\beta$-$Bi_2O_3$, when $\beta$-$Bi_2O_3$ and Cu-MOF form a heterojunction interface, electrons flow from the CB of Cu-MOF to the CB of $\beta$-$Bi_2O_3$ to reach the balanced state of the Fermi level [47]. This phenomenon is evidenced by the peak shift of Bi and the O shift in the XPS results (Figure 4b,c). The electron flow causes the interface region of $\beta$-$Bi_2O_3$ to become negatively charged and Cu-MOF to become positively charged, resulting in the $\beta$-$Bi_2O_3$ surface bending downward and the Cu-MOF surface bending upward [48]. Meanwhile, the space charge layer generated at the interface creates an internal potential gradient to oppose the electron flow, which facilitates the migration of photoinduced charge carriers [49].

The possible photocatalytic mechanism of the $Bi_2O_3$/MOF photocatalyst for 1,4-D degradation is proposed as shown in Figure 9. When irradiated by visible light, $\beta$-$Bi_2O_3$ and Cu-MOF absorb photons in response to visible light, and thus electrons ($e^-$) in VB are trans-

ferred from VB to CB, resulting in the formation of holes ($h^+$) in VB (Equations (4) and (5)). On the one hand, $e^-$ are consumed in the CBs under the action of dissolved oxygen ($O_2$) (Equation (6)) and the acidic environment (Equation (7)), which further inhibits electron–hole recombination. Importantly, oxidizing superoxide anion radicals ($\cdot O_2^-$) are generated from $O_2$ (−0.046 eV for $O_2/\cdot O_2^-$). On the other hand, the $h^+$ have enough potential to oxidize water molecules (2.37 eV for $H_2O/\cdot OH$), leading to the generation of hydroxyl radicals ($\cdot OH$) (Equation (8)) [50]. Therefore, $\cdot O_2^-$, $\cdot OH$, and $h^+$ collectively contribute to the degradation of 1,4-D.

$$Bi_2O_3 + hv \rightarrow e^- + h^+ \tag{4}$$

$$MOF + hv \rightarrow e^- + h^+ \tag{5}$$

$$O_2 + e^- \rightarrow H^+ + \bullet O_2^- \tag{6}$$

$$H^+ + e^- \rightarrow H_2 \tag{7}$$

$$H_2O + h^+ \rightarrow \bullet OH \tag{8}$$

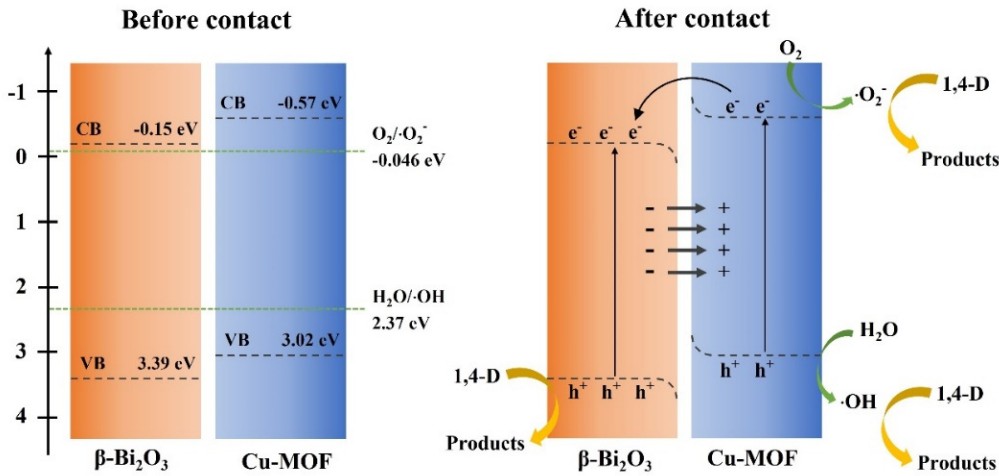

**Figure 9.** Proposed photocatalytic mechanism for the $Bi_2O_3/MOF$ photocatalyst.

## 3. Methods

### 3.1. Materials

All chemicals were of the highest purity and were used directly without further purification. 1,3,5-benzenetricarboxylic acid ($C_9H_6O_6$, $H_3BTC$), ethylene glycol (($CH_2OH$)$_2$, EG), polyethylenglycol ($C_{2n}H_{4n+2}O_{n+1}$, PEG), bismuth (III) nitrate pentahydrate ($Bi(NO_3)_3\cdot 5H_2O$), copper (II) nitrate trihydrate ($Cu(NO_3)_2\cdot 3H_2O$), and sodium borohydride ($NaBH_4$) were purchased from SHOWA (Tokyo, Japan). 2-propanol ($C_3H_8O$, IPA), ethanol ($C_2H_5OH$, EtOH), dimethyl sulfoxide ($C_2H_6OS$, DMSO), and Dimethylformamide ($C_3H_7NO$, DMF), 1, 4-D ($C_4H_8O_2$) were obtained from Acros (Morris Plains, NJ, USA). The 1,4-D stock solution (1000 mg/L) was prepared by dissolving 0.97 mL of 1,4-D in deionized (DI) water to yield a 1 L solution. DI water (Milli-Q Plus, resistance = 18.2 MΩ) was used for all experiments.

### 3.2. Synthesis of Composite Photocatalysts

The preparation processes of all photocatalysts are shown in Figure 10.

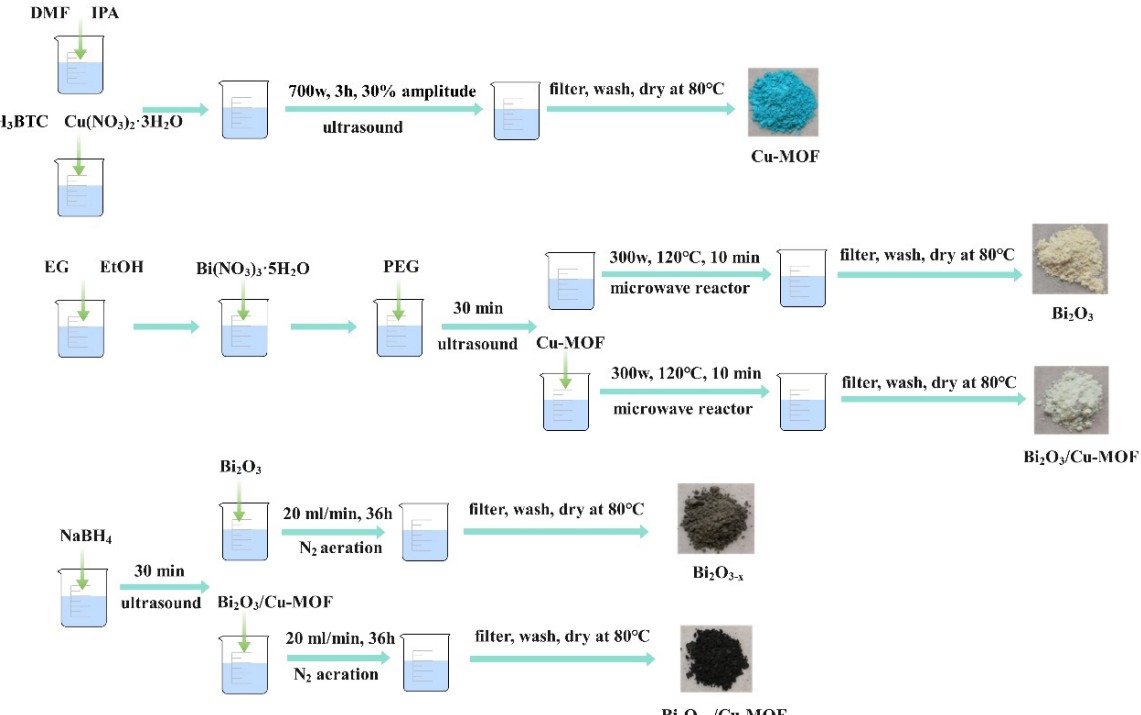

**Figure 10.** Preparation process of Cu-MOF, $Bi_2O_3$, $Bi_2O_3$/Cu-MOF, $Bi_2O_{3-x}$, and $Bi_2O_{3-x}$/Cu-MOF.

### 3.2.1. Cu-MOF

The Cu-MOF was synthesized using the ultrasonic method. Firstly, a 50 mL mixed solution was prepared by adding DMF, IPA, and DI water at a volume ratio of 1:1:1. Subsequently, 0.4200 g of $H_3BTC$ and 1.8301 g of $Cu(NO_3)_2 \cdot 3H_2O$ were dissolved in the mixed solution to obtain solution A and solution B, respectively. Then, solution B was added dropwise into solution A under continuous stirring for 30 min. Next, the resulting mixture was irradiated with high-intensity ultrasound (700 W, 20 kHz, Q700 SONICATOR) for 3 h at 30% amplitude. The ultrasound system followed a working cycle of 55 s on and 5 s off. The temperature was maintained at 15 °C by circulating water. Finally, the resultant product was filtered and washed three times with EtOH and water, and then dried overnight in a vacuum oven at 80 °C.

### 3.2.2. $Bi_2O_3$ and $Bi_2O_3$/Cu-MOF

The microwave method was used to synthesize $Bi_2O_3$ and Cu-MOF/$Bi_2O_3$. Firstly, 1.2662g $Bi(NO_3)_3 \cdot 5H_2O$ was dissolved in 15 mL of a mixture of EG and EtOH at a volume ratio of 1:3. Then, 0.0064 g of PEG was added to the solution, and the mixture was sonicated for 30 min to achieve complete reaction. Next, the obtained mixture was heated in a microwave reactor (Discover BenchMate system, CEM Co., Tokyo, Japan) with an operating power of 300 W and a temperature of 120 °C for 10 min. Finally, the formed precipitate was collected and washed several times with DI water and EtOH, then dried overnight in a vacuum oven at 80 °C, resulting in the obtained powder of $Bi_2O_3$. Cu-MOF/$Bi_2O_3$ was synthesized using the same procedure, with the addition of 0.0064 g of Cu-MOF before the microwave treatment.

### 3.2.3. $Bi_2O_{3-x}$ and $Bi_2O_{3-x}$/Cu-MOF

Firstly, 0.2600 g of $NaBH_4$ was added into 200 mL of DMSO and then sonicated for 30 min to completely dissolve it. A quantity of 2 g $Bi_2O_3$ was added to the mixture and stirring continued for 36 h under aeration $N_2$ (flow rate of 20 mL/min). Subsequently, the product was collected, washed several times with DI water and EtOH, and then dried overnight in a vacuum oven at 80 °C. The resulting product was denoted $Bi_2O_{3-x}$. To

synthesize Cu-MOF/Bi$_2$O$_{3-x}$, the same procedure was followed, but instead of adding Bi$_2$O$_3$, 2 g of Cu-MOF/Bi$_2$O$_3$ was used.

### 3.3. Characterizations

The morphologies of the catalysts were observed using a thermal field emission scanning electron microscope (FE-SEM, JEOL JSM-7800F, Tokyo, Japan) and transmission electron microscopy (TEM, JEM2010, JEOL, Tokyo, Japan). X-ray diffraction (XRD) patterns were recorded using a Rigaku Ultima III diffractometer (Tokyo, Japan) with Cu-K$\alpha$ radiation. The Brunauer–Emmett–Teller (BET) surface area, pore size, and pore volume of the catalysts were measured using the N$_2$ adsorption method using a Micrometrics ASAP-2020 nitrogen adsorption instrument. UV-visible diffuse reflectance spectra (DRS) were recorded using a Shimadzu UV-2600 spectrophotometer with an integrated sphere attachment, with barium sulfate as the reference. Photoluminescence (PL) properties were measured at room temperature using a Shimadzu RF-3501 spectrometer excited at 400 nm. The particle size distribution was determined using a Shimadzu SALD-2300 through laser diffraction scattering. X-ray photoelectron spectroscopy (XPS) was undertaken using a Physical Electronics PHI 5600 XPS instrument with monochromatic Al-Ka (1486.6 eV) as the excitation source.

### 3.4. Photocatalytic Activity

A quantity of 0.006 g of photocatalyst was dispersed in 100 mL of a 1,4-D (50 mg/L) solution. Prior to the photocatalytic experiment, the photocatalysts and solution were fully mixed in the dark for 30 min to reach the adsorption equilibrium. The reactor was then illuminated with a 350 W Xenon light (KIT-XENON-ADJ350W, Xenon arc, Bellevue, WA, USA) for 3 h. Samples were collected from the solution at regular intervals and filtered through a 0.22 μm membrane filter before analysis. All experiments were conducted at room temperature, and the solution pH was adjusted using HCl or NaOH. 1,4-D was analyzed using high-performance liquid chromatography (HPLC, LC-10 AT Shimadzu, Kyoto, Japan) equipped with an ultra aquo C18 column (5 μm, 250 mm × 4.6 mm) and a UV detection wavelength of 190 nm. The mobile phase consisted of 95% water and 5% acetonitrile, and the flow rate was set to 1 mL/min. The calibration curve was determined by establishing a linear relationship between known 1,4-D concentrations (0, 10, 20, 30, 40, and 50 mg/L) and HPLC peak areas.

## 4. Conclusions

In this study, the effects of Cu-MOF and oxygen vacancies on the improvement in the efficiency of 1,4-D degradation by Bi$_2$O$_3$ were investigated by preparing Bi$_2$O$_3$, Bi$_2$O$_3$/Cu-MOF, Bi$_2$O$_{3-x}$, and Bi$_2$O$_{3-x}$/Cu-MOF. The results showed that Bi$_2$O$_3$/Cu-MOF exhibited the highest photocatalytic activity for 1,4-D degradation. This superior performance can be attributed to several factors: the larger specific surface area, well-developed pore structure, and smaller particle size of Bi2O3/Cu-MOF, which provided a considerable number of active sites for catalysis. Meanwhile, the introduction of Cu-MOF enhanced the utilization efficiency of visible light and electron transfer, while the heterojunction promoted the effective separation of photogenerated electron–hole pairs. On the other hand, even though Bi$_2$O$_{3-x}$ and Bi$_2$O$_{3-x}$/Cu-MOF showed excellent optical properties, they surprisingly exhibited lower photocatalytic degradation efficiency of 1,4-D. This can be explained by the crystalline phase transition from β-Bi$_2$O$_3$ to α-Bi$_2$O$_3$, and the charge compensation effect, which reduced the number of oxidative holes (h$^+$) induced by oxygen vacancies. Therefore, the utilization of Cu-MOF for constructing heterojunctions significantly enhanced the efficiency of degradation of 1,4-D by Bi$_2$O$_3$, while oxygen vacancies and synergism of Cu-MOF and oxygen vacancies have negative effects. Bi$_2$O$_3$/Cu-MOF appears a promising photocatalysis for 1,4-D degradation.

**Author Contributions:** Conceptualization, Q.-Y.W. and J.J.W.; methodology, all authors; formal analysis, W.-M.W. and L.Z.; investigation, W.-M.W.; L.Z. and J.-Y.H.; writing—original draft preparation, W.-M.W.; writing—review editing and supervision, W.-L.W., Q.-Y.W. and J.J.W.; project administration, W.-M.W.; funding acquisition, Q.-Y.W. and J.J.W. All authors have read and agreed to the published version of the manuscript.

**Funding:** This research was funded by National Science and Technology Council (NSTC), Taiwan (NSTC-111-2221-E-035-017-MY3) and Shenzhen Science, Technology and Innovation Commission, China (Grant No. JCYJ20200109142829123).

**Data Availability Statement:** Data can be available upon request from the authors.

**Acknowledgments:** The authors wish to thank for the financial support by the Ministry of Science and Technology (MOST) in Taiwan under the contract numbers of NSTC-111-2221-E-035-017-MY3 and the Shenzhen Science, Technology and Innovation Commission (Grant No. JCYJ20200109142829123).

**Conflicts of Interest:** The authors declare no conflict of interest.

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
