# Peer review of "Photocatalytic Degradation of 1,4-Dioxane by Heterostructured Bi2O3/Cu-MOF Composites"

_catalysts, doi:10.3390/catal13081211_

Round 1
Author Response
The research article by Wang et al. designed Bi2O3/Cu-MOF composites for photocatalytic degradation of 1,4-Dioxane. This work provides good insights for the utilization of Cu-MOF and oxygen vacancy to enhance the 1,4-dioxane photocatalytic degradation efficiency of Bi2O3 by preparing different Cu-MOF and Bi2O3 composites. However, many unclear points and corrections should be fixed before considering it for publication.
- In line 105, the authors are suggested to rewrite the sentence ¨ The high magnification SEM images illustrated that the edges of microsphere structures were assembled from plentiful ultrathin nanosheets with a thickness of about 10 nm.¨
Reply: The sentence has been revised as “Further examination at high-magnification revealed that the peripheries of microsphere formations were constructed from abundant ultrathin nanosheets, each measuring approximately 10 nm in thickness” (Lines 103-105).
- The authors are suggested to explain the TEM images more deeply including fringe widths of different planes (obtained from XRD spectra) and also provide the EDX analysis.
Reply: Thanks for the suggestion. The primary objective of this manuscript is to investigate the effect of MOF and oxygen vacancies on the photocatalytic activity of Bi2O3. The XRD analysis is only intended to demonstrate whether the target materials are successfully synthesized in terms of crystallinity and crystalline phases. EDX has been conducted for demonstrating the main composition of Bi, C, Cu, and O without other impurities.
- The d-spacing values for various diffraction planes must be reported in XRD section.
Reply: The primary objective of this manuscript is to investigate the effect of MOF and oxygen vacancies on the photocatalytic activity of Bi2O3. The XRD analysis is only intended to demonstrate whether the target materials are successfully synthesized in terms of crystallinity and crystalline phases. Therefore, the d-space of the photocatalysts holds less relevance for the purpose of this study.
- In line 196, what is D50? The notation should be explained clearly for better understanding of text.
Reply: D50 represents “the median particle diameter”, which has been referred in the text (Line 190).
- The line 222 ¨ This phenomenon indicated that the oxygen vacancy could promote increasing the absorption intensity in the visible region.¨ must be rewritten for clear understanding.
Reply: The sentence has been revised as “This phenomenon elucidates that the presence of oxygen vacancies can enhance the absorption intensity within the visible spectral range” (Lines 214-216).
- The authors are suggested to provide FTIR analysis also for all the composites.
Reply: The XRD and XPS results can provide information about the crystal structure, chemical compositions, and surface chemical states of Bi2O3, Bi2O3/Cu-MOF, Bi2O3-x, and Bi2O3-x/Cu-MOF. These results offer essential insights into the materials. Therefore, FTIR spectroscopy was not taken into consideration for analyzing the surface functional groups of the materials.
- In figure 8 a, the axis title of x axis is missing.
Reply: Fig. 8a has the x axis title, which is “Time”.
- Authors must also provide the removal of 1,4-D in percentage as well.
Reply: We have provided the 1,4-D removal efficiency in the part of 2.2. “During the photocatalytic stage, the 1,4-D removal efficiency following the order of Bi2O3/Cu-MOF (68.7%) > Bi2O3 (62.8%) > Bi2O3-x/Cu-MOF (52.2%) >Bi2O3-x (47.1%) within 180 min.” (Line 266-267)
- The title ¨ 1,4-D photocatalytic degradation¨ must be corrected.
Reply: The title has been revised as “Photocatalytic activity” (Lines 389).
- Authors must provide a comparative analysis of reported photocatalytic activity with the previously reported materials.
Reply: Table 1 shows the 1,4-D removal efficiency of different photocatalysts. It was found that Bi2O3/Cu-MOF in this study exhibited higher photocatalytic activity for 1,4-D degradation compared with most of those previously reported (Lines 268-271)
Table 1. Comparison of photocatalytic activities of photocatalysts for 1,4-D.
|
Photocatalyst |
Light source |
Catalyst dosage (g/L) |
1,4-D concentration (mg/L) |
Time (min) |
1,4-D removal efficiency (%) |
references |
|
Fe/nAl |
solar |
0.3 |
50 |
180 |
about 22 |
[5] |
|
WO3/nγ-Al2O3 |
solar |
0.3 |
50 |
180 |
56.67 |
[43] |
|
TiO2 |
Xenon light (2 kW) |
0.5 |
500 |
180 |
about 10 |
[44] |
|
Au–TiO2 |
Xenon light (2 kW) |
0.5 |
500 |
240 |
59 |
[45] |
|
Cu-ZnO |
solar |
0.3 |
355 |
180 |
43.9 |
[46] |
|
Bi2O3/Cu-MOF |
Xenon light (350 W) |
0.06 |
50 |
180 |
68.7 |
This study |
Reviewer 2 Report
Dear Author,
I'm very glad to be the review of this paper (Manuscript Number: JWPE-D-23-01488). Title: Photocatalytic Degradation of 1,4-Dioxane by Heterostructured 2 Bi2O3/Cu-MOF Composites. I hope from the author of this paper take the following point in their consideration:
1. I could not find a recognizing contribution in this paper there are plenty of results, but the authors were not focusing on the novelty of the paper.
2. I have not found a comparative study with other catalyst such as SBA-15, MCM-41, MCM-48, and so on.
3. English should be improved throughout the manuscript.
4. The abstract must be rewritten again with a reduction and explaining the major finding with a conclusion
5. Re-write the introduction part to give more details.
6. The introduction part must be containing modern references such as:
Scientific Reports (2022) 12:16782. https://doi.org/10.1038/s41598-022-20984-0.
7. The aims of the present work must be more cleared at the end of the introduction part.
9. The surface area is very important for catalyst. The author must be discussing this point in detail because the surface area is a very important factor in increasing the reaction rate.
10. The author must be clear about the experimental steps for the preparation of calibration curves and stock solutions.
11. I hope the author draws the schematic diagram for all the processes.
12. Is the Hydrophilic or Hydrophobic catalyst performed in this study and Why?
13. What is the relationship between surface roughness and contact angle for the Cu-MOF, Bi2O3, Bi2O3/Cu-MOF, Bi2O3-x, Bi2O3- 124 x/Cu-MOF catalyst. The author must be discussing this point in detail.
14. The author must be clear about the experimental steps and discuss the interfacial phenomena for this study.
15. What is the main conclusion of this study? The conclusion must be reduced.
16. Maybe the leaching occurred from the surface of the catalyst Cu-MOF, Bi2O3, Bi2O3/Cu-MOF, Bi2O3-x, Bi2O3- 124 x/Cu-MOF materials that mean dissolved in the solution. I need the author to explain this important point in the result and discussion part.
17. The Cu-MOF, Bi2O3, Bi2O3/Cu-MOF, Bi2O3-x, Bi2O3- 124 x/Cu-MOF catalyst material characterizations must appear according to the sequence XRD, BET surface area, SEM, FT-IR, and TGA.
18. Why the author used a Cu-MOF, Bi2O3, Bi2O3/Cu-MOF, Bi2O3-x, Bi2O3- 124 x/Cu-MOF instead of other catalyst.
19. I hope the author of this manuscript achieves a comparison between this study and others the same reaction.
22. What is the purpose from Kinetics study such as; pseudo-first, second-order, and Intraparticle diffusion was included in this study.
23. The author must be clear the experimental steps for the preparation of catalyst and stock solution.
English should be improved throughout the manuscript.
Author Response
I'm very glad to be the review of this paper (Manuscript Number: JWPE-D-23-01488). Title: Photocatalytic Degradation of 1,4-Dioxane by Heterostructured 2 Bi2O3/Cu-MOF Composites. I hope from the author of this paper take the following point in their consideration:
- I could not find a recognizing contribution in this paper there are plenty of results, but the authors were not focusing on the novelty of the paper.
Reply: We have clarified the objectives and novelty in this paper (Lines 83-88).
- I have not found a comparative study with other catalyst such as SBA-15, MCM-41, MCM-48, and so on.
Reply: Table 1 shows the 1,4-D removal efficiency of different photocatalysts. It was found that Bi2O3/Cu-MOF in this study exhibited higher photocatalytic activity for 1,4-D degradation compared with most of those previously reported. (Lines 268-271)
Table 1. Comparison of photocatalytic activities of photocatalysts for 1,4-D.
|
Photocatalyst |
Light source |
Catalyst dosage (g/L) |
1,4-D concentration (mg/L) |
Time (min) |
1,4-D removal efficiency (%) |
references |
|
Fe/nAl |
solar |
0.3 |
50 |
180 |
about 22 |
[5] |
|
WO3/nγ-Al2O3 |
solar |
0.3 |
50 |
180 |
56.67 |
[43] |
|
TiO2 |
Xenon light (2 kW) |
0.5 |
500 |
180 |
about 10 |
[44] |
|
Au–TiO2 |
Xenon light (2 kW) |
0.5 |
500 |
240 |
59 |
[45] |
|
Cu-ZnO |
solar |
0.3 |
355 |
180 |
43.9 |
[46] |
|
Bi2O3/Cu-MOF |
Xenon light (350 W) |
0.06 |
50 |
180 |
68.7 |
This study |
- English should be improved throughout the manuscript.
Reply: We have conducted the proofreading throughout the manuscript.
- The abstract must be rewritten again with a reduction and explaining the major finding with a conclusion
Reply: We have revised the abstract and conclusions (Lines 27-31).
- Re-write the introduction part to give more details.
Reply: We have revised the introduction (Lines 83-88).
- The introduction part must be containing modern references such as:
Scientific Reports (2022) 12:16782. https://doi.org/10.1038/s41598-022-20984-0.
Reply: We have added the reference as indicated as a reference for this study (Line 53).
- The aims of the present work must be more cleared at the end of the introduction part.
Reply: We have revised the introduction (Lines 83-88).
- The surface area is very important for catalyst. The author must be discussing this point in detail because the surface area is a very important factor in increasing the reaction rate.
Reply: We have revised the discussion about surface area (Lines 182-186).
- The author must be clear about the experimental steps for the preparation of calibration curves and stock solutions.
Reply: We have added the experimental steps of calibration curves (Line 399-400) and stock solutions (Line 343-345).
- I hope the author draws the schematic diagram for all the processes.
Reply: The schematic diagram for preparation processes of Cu-MOF, Bi2O3, Bi2O3/Cu-MOF, Bi2O3-x, and Bi2O3-x/Cu-MOF is shown in Figure 10 (Lines 347-349).
- Is the Hydrophilic or Hydrophobic catalyst performed in this study and Why?
Reply: The main objective of this study is to investigate the photocatalytic activity of the photocatalysts, while the hydrophilicity and hydrophobicity of photocatalysts are not closely related to the purpose of this study. Therefore, the hydrophilicity and hydrophobicity of the photocatalysts were not considered in this study.
- What is the relationship between surface roughness and contact angle for the Cu-MOF, Bi2O3, Bi2O3/Cu-MOF, Bi2O3-x, Bi2O3-x/Cu-MOF catalyst. The author must be discussing this point in detail.
Reply: The primary purpose of this manuscript is to investigate the photocatalytic activity of the photocatalysts. Therelationship between surface roughness and contact angle of photocatalysts is not closely related to the purpose of this study. Therefore, it was not considered in this study.
- The author must be clear about the experimental steps and discuss the interfacial phenomena for this study.
Reply: The preparation processes of all catalysts are shown in Figure 10 (Lines 347-349), and detailed in section 3.2. As shown in Figure 10, Cu-MOF, Bi2O3, Bi2O3/Cu-MOF, Bi2O3-x, Bi2O3-x/Cu-MOF prepared in this study represented by different colors: blue, light yellow, white and grey black.
- What is the main conclusion of this study? The conclusion must be reduced.
Reply: We have revised the conclusion (Lines 413-416).
- Maybe the leaching occurred from the surface of the catalyst Cu-MOF, Bi2O3, Bi2O3/Cu-MOF, Bi2O3-x, Bi2O3-x/Cu-MOF materials that mean dissolved in the solution. I need the author to explain this important point in the result and discussion part.
Reply: We ever conducted the leaching experiments. The prepared materials were immersed in DI water for 180 min, then filtered, and the leachate was used for photocatalytic experiments. 1,4-D was hardly degraded in this system, so we think that the effect of dissolving the materials in water does not need to be considered.
- The Cu-MOF, Bi2O3, Bi2O3/Cu-MOF, Bi2O3-x, Bi2O3-x/Cu-MOF catalyst material characterizations must appear according to the sequence XRD, BET surface area, SEM, FT-IR, and TGA.
Reply: In this study, we used to observe the whole morphology and structure of the materials first (SEM, TEM), then investigate the physicochemical properties of the materials sequentially (XRD, XPS, BET, D50), and finally explore their optical properties (UV-VIS, PL). From these, the basic information of the materials can be obtained. Therefore, FTIR and TGA was not taken into consideration.
- Why the author used a Cu-MOF, Bi2O3, Bi2O3/Cu-MOF, Bi2O3-x, Bi2O3-x/Cu-MOF instead of other catalyst.
Reply: We have already elucidated the reasons for the selection of these catalysts in the introduction section (Lines 52-88).
- I hope the author of this manuscript achieves a comparison between this study and others the same reaction.
Reply: Table 1 shows the 1,4-D removal efficiency of different photocatalysts. It was found that Bi2O3/Cu-MOF in this study exhibited higher photocatalytic activity for 1,4-D degradation compared with most of those previously reported (Lines 265-268).
Table 1. Comparison of photocatalytic activities of photocatalysts for 1,4-D.
|
Photocatalyst |
Light source |
Catalyst dosage (g/L) |
1,4-D concentration (mg/L) |
Time (min) |
1,4-D removal efficiency (%) |
references |
|
Fe/nAl |
solar |
0.3 |
50 |
180 |
about 22 |
[5] |
|
WO3/nγ-Al2O3 |
solar |
0.3 |
50 |
180 |
56.67 |
[43] |
|
TiO2 |
Xenon light (2 kW) |
0.5 |
500 |
180 |
about 10 |
[44] |
|
Au–TiO2 |
Xenon light (2 kW) |
0.5 |
500 |
240 |
59 |
[45] |
|
Cu-ZnO |
solar |
0.3 |
355 |
180 |
43.9 |
[46] |
|
Bi2O3/Cu-MOF |
Xenon light (350 W) |
0.06 |
50 |
180 |
68.7 |
This study |
- What is the purpose from Kinetics study such as; pseudo-first, second-order, and Intraparticle diffusion was included in this study.
Reply: According to the previous studies, the degradation process of organic matter can usually be better fitted by pseudo-first-order kinetics. In this study, the kobs calculated by pseudo-first-order kinetics were used to visualize the degradation rate of 1,4-D by different photocatalysts.
- The author must be clear the experimental steps for the preparation of catalyst and stock solution.
Reply: The preparation processes of all catalysts are shown in Figure 10 (Lines 347-349) and as detailed in section 3.2. We have also added the experimental steps of stock solutions (Lines 343-345).
Round 2
Reviewer 2 Report
Thank you very much. I decided to accept this paper for publication because the author was achieved all comments in detail.